

# The expression characteristics and prognostic roles of autophagy-related genes in gastric cancer

Mengya Wang[1,2,*], Jingjing Jing[1,2,*], Hao Li[3], Jingwei Liu[2], Yuan Yuan[1,2] and Liping Sun[1,2]

[1] Tumor Etiology and Screening Department of Cancer Institute, and Key Laboratory of Cancer Etiology and Prevention in Liaoning Education Department, the First Hospital of China Medical University, Shenyang, China
[2] Key Laboratory of GI Cancer Etiology and Prevention in Liaoning Province, the First Hospital of China Medical University, Shenyang, China
[3] Department of Clinical Laboratory, the First Hospital of China Medical University, Shenyang, China
[*] These authors contributed equally to this work.

Corresponding authors
Yuan Yuan, yyuansubmission@hotmail.com
Liping Sun, lpsun@cmu.edu.cn

## ABSTRACT

**Background**. Autophagy is an evolutionarily highly conserved process, accompanied by the dynamic changes of various molecules, which is necessary for the orderly degradation and recycling of cellular components. The aim of the study was to identify the role of autophagy-related (*ATG*) genes in the occurrence and development of gastric cancer (GC).

**Methods**. Data from Oncomine dataset was used for the differential expression analysis between cancer and normal tissues. The association of *ATG* genes expression with clinicopathologic indicators was evaluated by The Cancer Genome Atlas (TCGA) database and Gene Expression Omnibus (GEO) database. Moreover, using the TCGA datasets, the prognostic role of *ATG* genes was assessed. A nomogram was further built to assess the independent prognostic factors.

**Results**. The expression of autophagy-related genes *AMBRA1*, *ATG4B*, *ATG7*, *ATG10*, *ATG12*, *ATG16L2*, *GABARAPL2*, *GABARAPL1*, *ULK4* and *WIPI2* showed differences between cancer and normal tissues. After verification, *ATG14* and *ATG4D* were significantly associated with TNM stage. *ATG9A*, *ATG2A*, and *ATG4D* were associated with T stage. *VMP1* and *ATG4A* were low-expressed in patients without lymph node metastasis. No gene in autophagy pathway was associated with M stage. Further multivariate analysis suggested that *ATG4D* and *MAP1LC3C* were independent prognostic factors for GC. The C-index of nomogram was 0.676 and the 95% CI was 0.628 to 0.724.

**Conclusion**. Our study provided a comprehensive illustration of *ATG* genes expression characteristics in GC. Abnormal expressions of the ubiquitin-like conjugated system in *ATG* genes plays a key role in the occurrence of GC. *ATG8/LC3* sub-system may play an important role in development and clinical outcome of GC. In the future, it is necessary to further elucidate the alterations of specific *ATG8/LC3* forms in order to provide insights for the discovery, diagnosis, or targeting for GC.

## INTRODUCTION

Autophagy is an evolutionally highly conserved process, which is necessary for the orderly degradation and recycling of cellular components (*Yang et al., 2020*). In normal cells, autophagy keeps low-level constitutive function. Basal autophagy plays an important role in maintaining homeostatic control and elimination of unfavorable proteins. Its activity can be accelerated by a variety of cellular stressors including nutrient starvation, DNA damage, and organelle damage. Autophagy is closely related to the occurrence and treatment of tumors (*Rahman et al., 2020*). Recently, the paradoxical roles of autophagy in tumor suppression and tumor promotion have been widely observed. As a physiological quality control process, autophagy exerts a cytoprotective effect to suppress cancer development by removing damage that leads to aberrant mutations. On the other hand, as cancer progresses, starving and oxidative stress situation can active autophagy to fulfill the high metabolic need of cancer cells (*Mathew, Karantza-Wadsworth & White, 2007*).

The process of autophagy is accompanied by the dynamic changes of various molecules. Identification of the autophagy-related biomarkers will contribute to improving diagnosis and treatment of cancers. Autophagy is executed by a set of autophagy-related (*ATG*) genes, which have been investigated extensively in yeast. Although the discovery of *ATG* genes greatly advanced the understanding of autophagy, the function and mechanisms involved in *ATG* genes need to be further explored in mammalian. Recently, several studies have investigated the association of *ATG* genes and cancers. By activating *ATG6*-mediated autophagy, the down-regulation of microRNA-30a increases the chemoresistance of osteosarcoma cells, thereby inhibiting cell proliferation and invasion (*Xu et al., 2016*). Upregulation of UCA1 inhibits cell proliferation, migration, invasion, and drug resistance via *ATG7*-mediated autophagy (*Wu et al., 2019*). The methyltransferase MGMT inhibits the expression of *ATG4B*, thereby inhibiting autophagy and reducing the chemosensitivity of cisplatin in gastric cancer (GC) (*Lei et al., 2020*). Moreover, comprehensive study of all *ATG* genes has been conducted in breast, head neck and kidney carcinoma (*Deng et al., 2018*; *Pei et al., 2018*).

GC is the fourth most common cancer and the second leading cause of cancer death in the world (*Van Cutsem et al., 2016*). The incidence is mainly related to diet, lifestyle, genetic predisposition, family history, treatment and medical conditions, infections, demographic characteristics, occupational exposures and ionizing radiation (*Yusefi et al., 2018*). Abnormal expression of *ATG* gen*es* may lead to the dysregulation of autophagy and tumorigenesis. However, the diagnostic and prognostic values of *ATG* genes have not been fully realized in GC. Since large-scale expression data is available, it is feasible to display an overview of *ATG* genes from the perspective of expression characteristics and prognostic role in GC. In the current study, we performed systematic analysis by using available datasets of ONCOMINE and The Cancer Genome Atlas (TCGA), in order to evaluate the differential expression of *ATG* genes and their associations with clinicopathological parameters and prognosis of GC. Our data may provide a new understanding of the autophagy-related mechanism in gastric carcinogenesis.

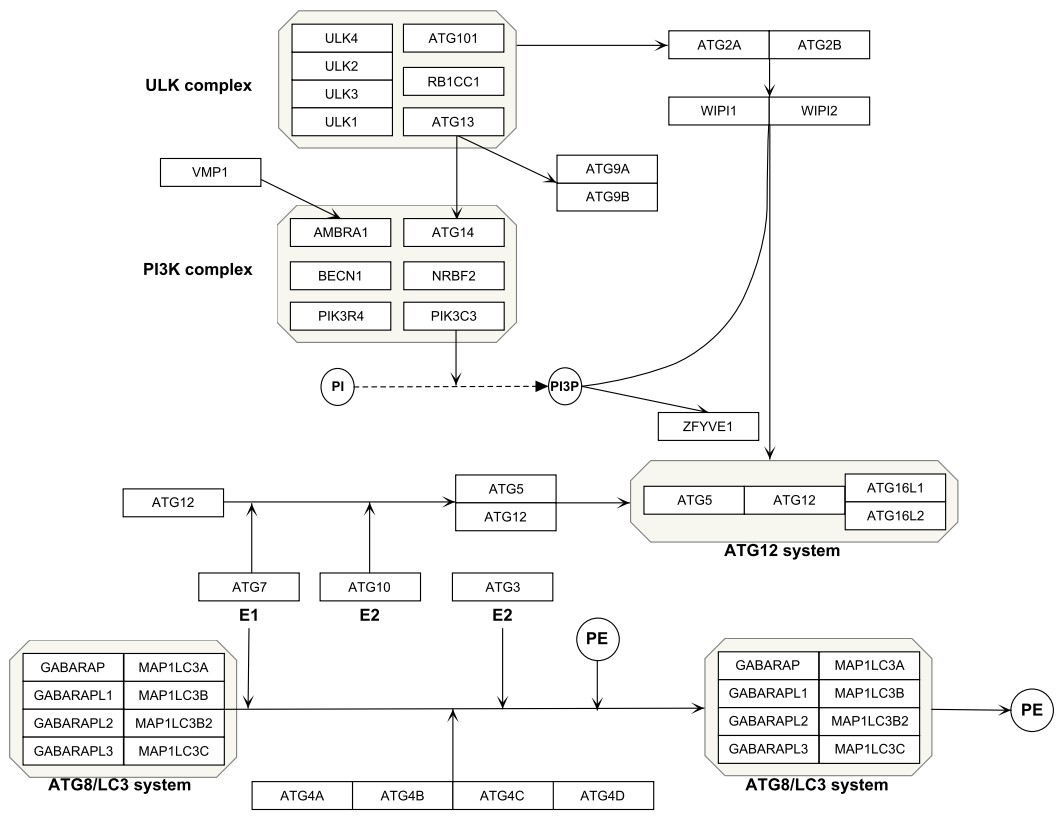

**Figure 1   Schematic of autophagy pathway.**

## MATERIALS & METHODS

### Autophagy-related genes selection

The Kyoto Encyclopedia of Genes and Genomes (KEGG, https://www.kegg.jp/) is an   online tool for analysis of the gene function (*Kanehisa et al., 2020*). Reactome (https://reactome.org/) is a bioinformatics resource for visualization, interpretation and analysis of pathways (*Jassal et al., 2020*). Using the two datasets, we selected the genes in autophagy pathways as *ATG* genes, which composed four functional units including the *ULK* protein complex, *Beclin-1/PI3K* complex, ubiquitin-like conjugation system and other genes (*Mizushima, Yoshimori & Ohsumi, 2011*). All the isoforms of a gene were included, such as *ATG4A*, *ATG4B*, *ATG4C* and *ATG4D*. A total of 40 genes were selected. PathVisio (Version:3.3.0, https://pathvisio.github.io/) was used to visualize the autophagy genes (*Kutmon et al., 2015*). which composed four functional units including *ULK* complex, *PI3K* complex, ubiquitin-like conjugation system and other genes (Table 1 and Fig. 1).

### Differential gene expression analysis by Oncomine

By consulting the public data in Oncomine (https://www.oncomine.org/resource/login. html) (*Rhodes et al., 2004*), a powerful online database with 715 sub datasets and 86,733 samples, we established and logged in an Oncomine account, and input all of 40 *ATG* genes

**Table 1** Description of autophagy related gene.

| | Gene symbol | Aliases | Function |
|---|---|---|---|
| ULK complex | ULK1/2/3/4 | ATG1A/B/C/D | Acts upstream of PIK3C3 to regulate the formation of autophagophores |
| | ATG101 | C12orf44 | Stabilizes ATG13, protecting it from proteasomal degradation. |
| | ATG13 | KIAA0652 | Essential for autophagosome formation |
| | RB1CC1 | ATG17 | Direct interaction with Atg16L1 |
| PI3K complex | BECN1 | ATG6 | Acts as core subunit of the PI3K complex |
| | PIK3R4 | VPS15 | Involved in regulation of degradative endocytic trafficking |
| | PIK3C3 | VPS34 | Catalytic subunit of the PI3K complex |
| | NRBF2 | COPR | Modulated ATG14 protein |
| | ATG14 | ATG14L | Plays a role in autophagosome formation and MAP1LC3/LC3 conjugation to phosphatidylethanolamine |
| | AMBRA1 | DCAF3 | Interacts with becn1 |
| ubiquitin-like conjugating system | ATG12 | APG12 | Conjugation with ATG5 |
| | ATG5 | APG5 | Functions as an E1-like activating enzyme |
| | ATG16L1/L2 | ATG16A/B | Interacts with ATG12-ATG5 to mediate the conjugation of phosphatidylethanolamine (PE) to LC3 |
| | ATG3 | APG3 | E2 conjugating enzyme |
| | ATG4A/B/C/D | APG4A/B/C/D | Cleaves the C-terminal amino acid of ATG8 family proteins to reveal a C-terminal glycine |
| | ATG7 | APG7 | E1-like activating enzyme |
| | ATG10 | APG10 | E2-like enzyme |
| | GABARAP/L1/L2/L3 | ATG8A/B/C/D | Ubiquitin-like modifier |
| | MAP1LC3A/B/B2/C | ATG8E/F/G/J | Ubiquitin-like modifier |
| others | WIPI1/2 | ATG18A/B | Functions upstream of the ATG12-ATG5-ATG16L1 complex and LC3, and downstream of the ULK1 and PI3-kinase complexes |
| | ATG9A/B | APG9L1/L2 | Transmembrane protein |
| | ATG2A/B | / | Required for both autophagosome formation |
| | ZFYVE1 | DFCP1 | PI3P-binding FYVE-containing protein |
| | VMP1 | EPG3, TANGO5, TMEM49 | Plays a role in the initial stages of the autophagic process through its interaction with BECN1 |

(gene symbols were listed in Table 1) to analyze their differential expression at transcription level in GC and different Lauren types. Combination of $P$-value <0.05 and fold change >2 was identified as significant difference in gene expression.

## Correlation analysis of *ATG* genes expression and clinicopathological parameters from TCGA and GEO datasets

TCGA is a public database that contains the data of genomic expressions and the clinical features in 33 types of cancers (*Tomczak, Czerwinska & Wiznerowicz, 2015*). The gene expression and clinicopathological information of GC were downloaded from TCGA data portal (https://portal.gdc.cancer.gov/projects/). R was performed to normalize the expression data. The patients' TNM stage, T, N and M (*Nagtegaal et al., 2020*) were considered as the clinical parameters.

### Verification of the differences of gene expression

The GSE62254 dataset was a 300 samples microarray profile tested by the Asian Cancer Research Group (ACRG) (Cristescu et al., 2015). Using GSE62254, we verified the differences of gene expression identified from TCGA datasets.

### Statistical analysis

All statistical analyses were performed by R 3.14 (http://www.r-project.org/) and the package of rms. Student's $t$-tests was used to analyze the differences between cancer samples and normal tissues, of which the criterion is $p$-value $<0.01$ and fold change $>2.0$. The association between the $ATG$ genes expressions and clinical features was accessed by Pearson $X^2$ test. The correlation between $ATG$ genes expressions and overall survival time was evaluated by Kaplan–Meier method and compared by log-rank test. Univariate and multivariate Cox proportional hazard regression models were used to recognize the independent prognostic factors. Based on the multivariate Cox regression models, a nomogram was formulated together with all the independent prognostic genes. The concordance index (C-index), which is similar to the area under the receiver operating characteristic (ROC), was used to evaluate the nomogram. $P < 0.05$ were considered significant difference.

## RESULTS

### Differential expression of *ATG* genes in GC

By the Oncomine analysis, there were 10 genes of 40 *ATG* genes with significantly differential expression between GC and normal samples, which were named as differentially expressed genes (DEGs) (Fig. 2). Seven DEGs were belong to the ubiquitin-like conjugating system, among them *ATG4B*, *ATG12* and *ATG16L2* were significantly up-regulated in GC, while *ATG10*, *GABARAPL2* and *GABARAPL1* expressions were down-regulated in GC. As for *ATG7*, the expression was uncertain. *ULK4*, belonging to the *ULK* complex, was found down-regulated in GC. While *AMBRA1*, a member of the *PI3K* complex, was highly expressed in GC. As a connection between *PI3K* complex, *ULK* complex and *ATG12* system, *WIPI2* showed higher expression in cancer tissue.

Histological stratification analysis showed that *GABARAPL1* was down-regulated in all types of GC compared with normal tissues, with fold change of $-2.321$ in intestinal gastric adenocarcinoma, $-2.287$ fold in diffuse adenocarcinoma and $-2.622$ fold in mixed adenocarcinoma. Six DEGs showed significant differences in the gastric mixed adenocarcinoma subgroup, among them *AMBRA1*, *ATG4B*, *ATG7* (probe 224025_s_at) and *ATG12* were up-regulated, while *GABARAPL1* and *ATG7* (probe 1569827_at) were down-regulated. Four DEGs including *ATG10*, *ATG16L2*, *ULK4* and *GABARAPL1* showed differences in diffuse gastric adenocarcinoma subgroup, while other four DEGs including *ATG7* (probe 224025_s_at), *GABARAPL1*, *WIPI2* and *GABARAPL3* showed differences in gastric intestinal type adenocarcinoma subgroup (Figs. 3A and 3B).

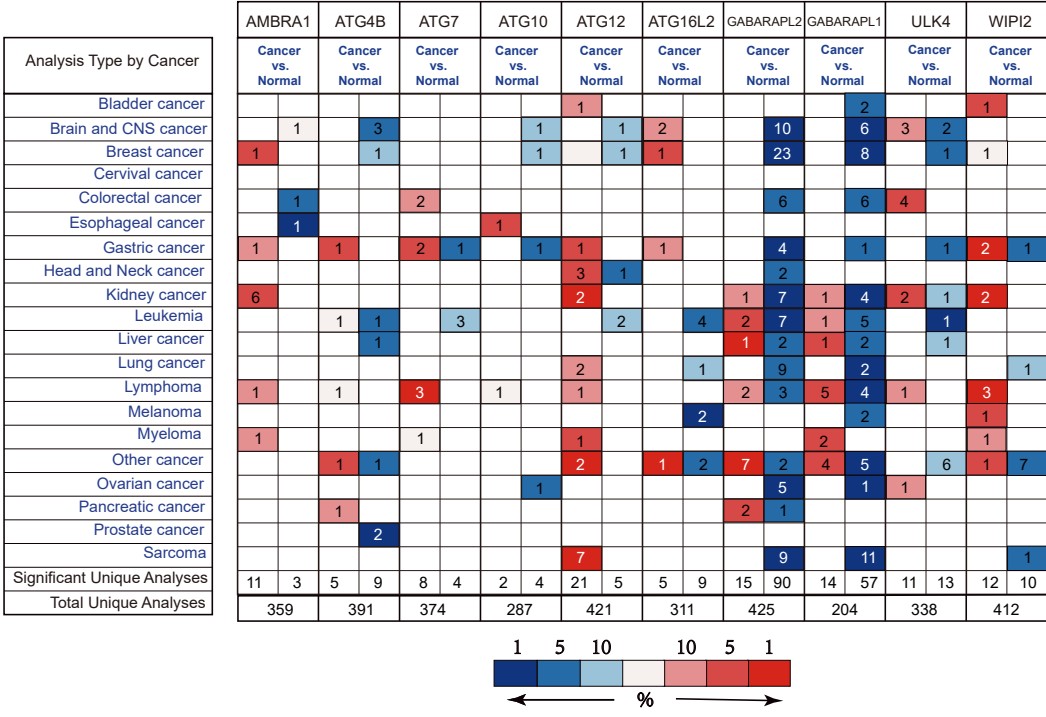

**Figure 2** **Different ATGs mRNA expression in different tumor types.** This graphic showed the numbers of datasets with statistically signifcant mRNA overexpression (red) or downexpression (blue) of the target gene (cancer vs. normal tissue). Cell color is determined by the best gene rank percentile for the analyses within the cell.

## Association between *ATG* genes expression and clinicopathologic variables of GC

Data of 376 GC patients in TCGA were downloaded for the analysis. *ATG14*, *ULK3*, *ATG2B*, *ATG12*, *ATG4C*, *ATG4D*, and *MAP1LC3A* showed significantly relationship with TNM stage. After verification, *ATG14* and *ATG4D* were significantly associated with TNM stage ($P = 0.027, 0.048$ respectively). *ATG9A* ($P = 0.00083$), *ATG2A* ($P = 0.00417$), and *ATG4D* ($P = 0.00864$) were related with T stage. Low expression of *VMP1* and *ATG4A* suggested absence of lymph node metastasis ($P = 0.0018, 0.015$, correspondingly). However, no gene in autophagy pathway was observed to be associated with M stage (Table 2).

## Roles of *ATG* genes expression in the prediction of GC prognosis

354 patients were included to analyze the overall survival of GC. The median value was used as cut-off value to distinguish high expression and low expression of *ATG* genes. According to the univariate survival analysis, *ATG4D*, *GABARAPL2* and *MAP1LC3C* were significantly associated with the prognosis of GC. Moreover, the patients with low-expression of *ATG4D* or high-expression of *GABARAPL2* and *MAP1LC3C* demonstrated longer survival time, and both of the latter two genes belonged to *ATG8/LC3* system. Using the Cox's proportional hazards model, we then performed the multivariate analysis adjusted by gender, age, TNM stage. *ATG4D* and *MAP1LC3C* were identified as the independent

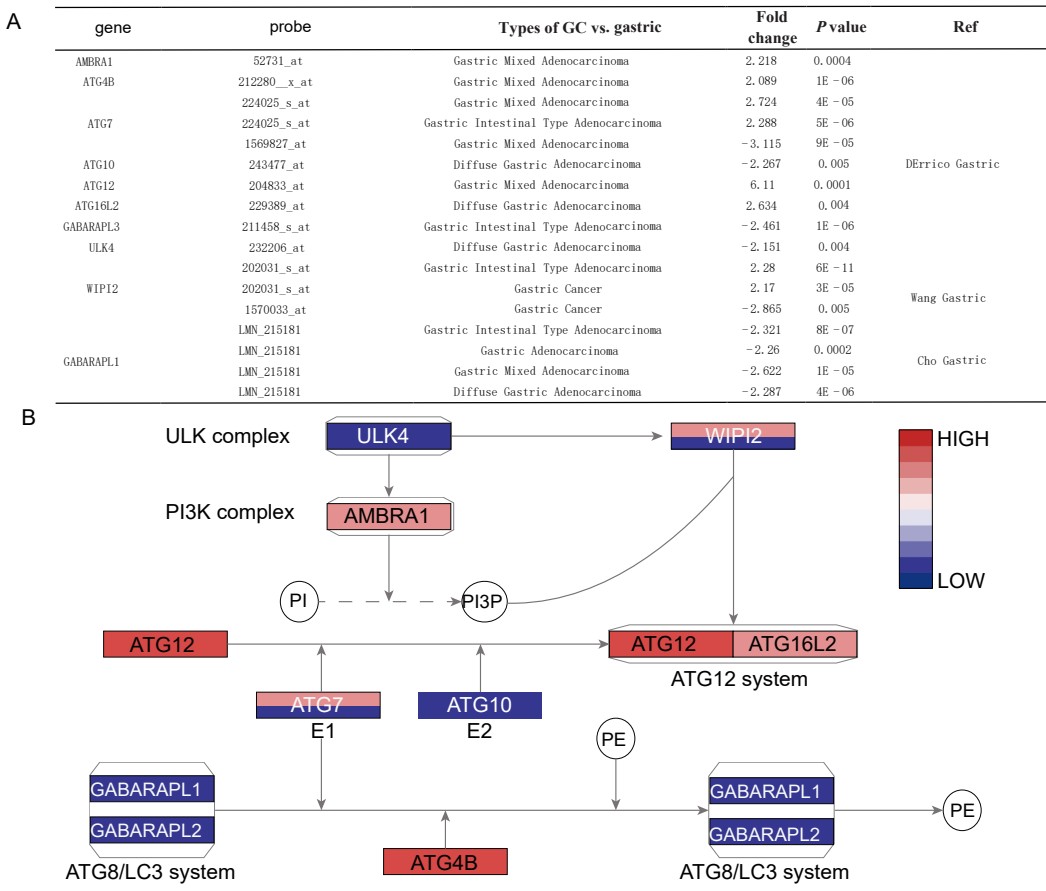

**A**

| gene | probe | Types of GC vs. gastric | Fold change | P value | Ref |
|---|---|---|---|---|---|
| AMBRA1 | 52731_at | Gastric Mixed Adenocarcinoma | 2.218 | 0.0004 | |
| ATG4B | 212280__x_at | Gastric Mixed Adenocarcinoma | 2.089 | 1E−06 | |
| | 224025_s_at | Gastric Mixed Adenocarcinoma | 2.724 | 4E−05 | |
| ATG7 | 224025_s_at | Gastric Intestinal Type Adenocarcinoma | 2.288 | 5E−06 | |
| | 1569827_at | Gastric Mixed Adenocarcinoma | −3.115 | 9E−05 | |
| ATG10 | 243477_at | Diffuse Gastric Adenocarcinoma | −2.267 | 0.005 | DErrico Gastric |
| ATG12 | 204833_at | Gastric Mixed Adenocarcinoma | 6.11 | 0.0001 | |
| ATG16L2 | 229389_at | Diffuse Gastric Adenocarcinoma | 2.634 | 0.004 | |
| GABARAPL3 | 211458_s_at | Gastric Intestinal Type Adenocarcinoma | −2.461 | 1E−06 | |
| ULK4 | 232206_at | Diffuse Gastric Adenocarcinoma | −2.151 | 0.004 | |
| | 202031_s_at | Gastric Intestinal Type Adenocarcinoma | 2.28 | 6E−11 | |
| WIPI2 | 202031_s_at | Gastric Cancer | 2.17 | 3E−05 | Wang Gastric |
| | 1570033_at | Gastric Cancer | −2.865 | 0.005 | |
| | LMN_215181 | Gastric Intestinal Type Adenocarcinoma | −2.321 | 8E−07 | |
| GABARAPL1 | LMN_215181 | Gastric Adenocarcinoma | −2.26 | 0.0002 | Cho Gastric |
| | LMN_215181 | Gastric Mixed Adenocarcinoma | −2.622 | 1E−05 | |
| | LMN_215181 | Diffuse Gastric Adenocarcinoma | −2.287 | 4E−06 | |

**Figure 3 The detail information for the different ATGs.** (A) The detail information in the oncomine dataset. (B) The position of different ATGs in autophagy pathway. The blue color represents downexpression in cancer, while the red color represents overexpression in cancer. The gene with two different colors means two probe of the gene showed different expressions. The gradient color represents the gene's fold change.

prognostic factors, with adjusted hazard ratio (HR) of 1.5727 (95% CI [1.1194–2.21]) and 0.5767 (95% CI [0.4086–0.8138]) separately (Fig. 4 and Table 3). The summary of the correlation between *ATG* genes expression and TNM staging and prognosis of GC was shown in Fig. 5.

## Joint prediction of the GC prognosis using *ATG4D* and *MAP1LC3C*

According to the expression of *ATG4D* and *MAP1LC3C* in GC, the gastric cancer patients were divided into four groups: *ATG4D* high expression - *MAP1LC3C* high expression (HH), *ATG4D* low expression - *MAP1LC3C* low expression (LL), *ATG4D* high expression - *MAP1LC3C* low expression (HL) and *ATG4D* low expression - *MAP1LC3C* high expression (LH). A significant difference was displayed among the four groups ($p = 0.0056$, Fig. 6A).

Furtherly, to predict 1-year and 3-year survival rate, we built a nomogram by the multivariate Cox regression models. After validation, the C-index was 0.676 and the 95% CI was 0.628 to 0.724. According to the total score after added with points identified on

**Table 2 The association between autophagy related gene and TNM stage.**

| Gene symbol | | TCGA | | | GSE62254 | | |
|---|---|---|---|---|---|---|---|
| | | **TNM** | | | | | |
| | | **I–II** | **III–IV** | **P** | **I–II** | **III–IV** | **P** |
| ATG14 | low | 96 | 80 | | 72 | 76 | |
| | high | 71 | 105 | 0.00761 | 54 | 96 | **0.02711** |
| ULK3 | low | 71 | 100 | | 61 | 89 | |
| | high | 96 | 85 | 0.03 | 65 | 83 | 0.568 |
| ATG2B | low | 95 | 85 | | 62 | 86 | |
| | high | 72 | 100 | 0.04032 | 64 | 86 | 0.8923 |
| ATG12 | low | 96 | 83 | | 58 | 91 | |
| | high | 71 | 102 | 0.018 | 68 | 81 | 0.241 |
| ATG4C | low | 94 | 84 | | 65 | 83 | |
| | high | 73 | 101 | 0.041 | 61 | 89 | 0.57 |
| ATG4D | low | 70 | 100 | | 55 | 95 | |
| | high | 97 | 85 | 0.022 | 71 | 77 | **0.04822** |
| MAP1LC3A | low | 71 | 105 | | 65 | 83 | |
| | high | 96 | 80 | 0.0076 | 61 | 89 | 0.57 |
| | | **T** | | | | | |
| | | **T1T2** | **T3T4** | **P** | **T1T2** | **T3T4** | **P** |
| WIPI1 | low | 41 | 147 | | 101 | 49 | |
| | high | 58 | 121 | 0.022 | 87 | 63 | 0.0947 |
| ATG9A | low | 39 | 143 | | 80 | 70 | |
| | high | 60 | 125 | 0.018 | 108 | 42 | **0.00083** |
| ATG2B | low | 42 | 145 | | 95 | 85 | |
| | high | 57 | 123 | 0.047 | 93 | 57 | 0.092 |
| ATG2A | low | 40 | 141 | | 82 | 68 | |
| | high | 59 | 127 | 0.038 | 106 | 44 | **0.00417** |
| ATG4D | low | 39 | 141 | | 83 | 67 | |
| | high | 60 | 127 | 0.025 | 105 | 45 | **0.00864** |
| ATG7 | low | 40 | 146 | | 90 | 60 | |
| | high | 59 | 122 | 0.017 | 98 | 52 | 0.3396 |
| | | **N** | | | | | |
| | | **N0** | **!N0** | **P** | **N0** | **!N0** | **P** |
| PIK3R4 | low | 66 | 116 | | 18 | 132 | |
| | high | 45 | 130 | 0.031 | 20 | 130 | 0.728 |
| VMP1 | low | 65 | 116 | | 28 | 122 | |
| | high | 46 | 130 | 0.046 | 10 | 140 | **0.00178** |
| ATG12 | low | 66 | 113 | | 16 | 134 | |
| | high | 45 | 133 | 0.018 | 22 | 128 | 0.2976 |
| ATG4A | low | 68 | 115 | | 26 | 124 | |
| | high | 43 | 131 | 0.0111 | 12 | 138 | **0.01509** |

**Table 2** (*continued*)

| Gene symbol | | TCGA | | | GSE62254 | | |
|---|---|---|---|---|---|---|---|
| | | M | | | M | | |
| | | M0 | M1 | P | M0 | M1 | P |
| ULK4 | low | 160 | 19 | | 137 | 13 | |
| | high | 170 | 6 | **0.008** | 136 | 14 | 0.8401 |
| MAP1LC3B | low | 174 | 7 | | 139 | 11 | |
| | high | 156 | 18 | **0.0171** | 134 | 16 | 0.3131 |

**Notes.**
  Significant results are marked in bold.

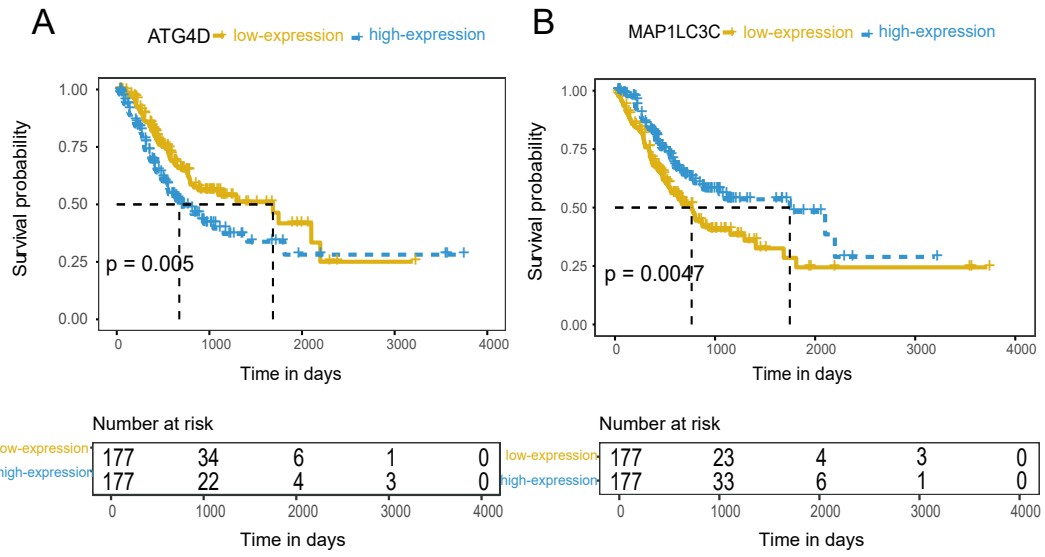

**Figure 4** **The prognostic value of mRNA level of independent prognostic factors.** (A) ATG4D. (B) MAP1LC3C.

the point scale, we found that the likelihood of 1-year and 3-year OS for individual patient could be reasonably predicated by nomogram (Fig. 6B). As shown in Figs. 6C and 6D, the survival evaluated by the Kaplan–Meier method was marked on the y-axes, the predicted survival estimated by nomogram was observed on the x-axes, and the red lines represented the ideal reference line for which predicted survival corresponds with actual survival. The plot for the probability of OS 1-year or 3-year showed optimal agreement between the prediction by nomogram and actual observation for nomogram.

## DISCUSSION

Considering the vital function of *ATG* genes in autophagy, many studies have focused on their role in cancers. To date, no researcher has elaborated an overview of the impact of *ATG* genes on the development, progression, and prognosis of GC. In the current study, for the first time, we investigated the expression profiling and the prognostic roles of whole members of *ATG* genes in GC using multiple databases. Our results elucidated

**Table 3  Prognosis analysis of autophagy related gene in TCGA datasets.**

|  | Univirable analysis | | Multivanable analysis | |
|---|---|---|---|---|
|  | **HR(95CI)** | **P** | **HR(95CI)** | **P** |
| ATG4D | 1.602(1.153–2.225) | **0.00493** | 1.5727(1.1194–2.21) | **0.009058** |
| GABARAPL2 | 0.6925(0.499–0.9609) | **0.0279** | 0.7855(0.5597–1.102) | 0.162447 |
| MAP1LC3C | 0.6242(0.4488–0.8682) | **0.00511** | 0.5767(0.4086–0.8138) | **0.00173** |

**Notes.**
Significant results are marked in bold.

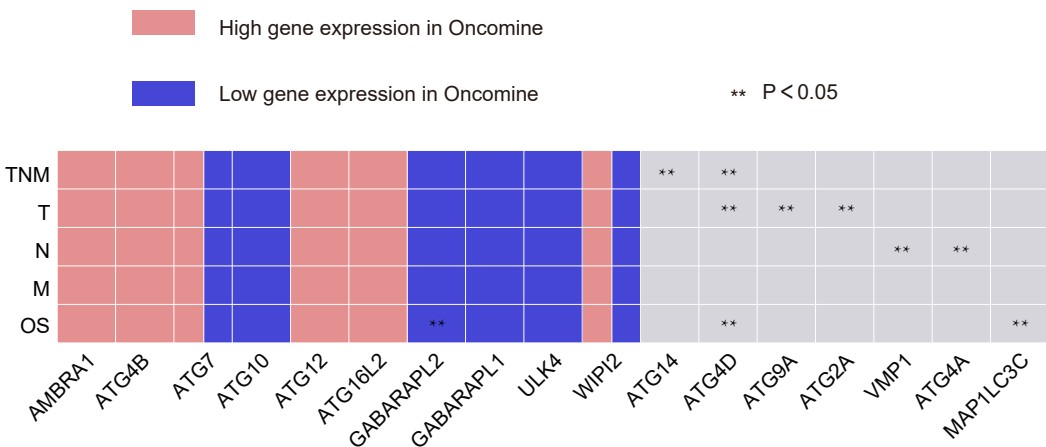

**Figure 5  Summary of the correlation between *ATG* genes expression and TNM staging and prognosis.** The red frame represents genes with high significant expression, and the blue frame represents genes with low significant expression. Two asterisks (**) represent that gene expressions has significant correlation with TNM staging or OS of GC.

that abnormal expressions of some key *ATG* genes were significantly associated with GC progression and outcome.

Firstly, 10 DEGs were identified between cancer and normal tissues, and 7 of these genes were associated with ubiquitin-like conjugating system, which intimately involved in driving the biogenesis of the autophagosomal membrane (*Nakatogawa, 2013*). *ATG4B* (*Liu et al., 2014*), the core autophagy protein in *ATG8/LC3* system, was found to be up-regulated in cancer tissue in our study. It has been reported that *ATG4B* can promote the growth of colorectal cancer, while silencing the expression of *ATG4B* can reduce the colony formation of cancer cells and inhibit tumor growth (*Liu et al., 2014*; *Liu et al., 2018*). The E1-like activating enzyme *ATG7* and the E2-like activating enzyme *ATG10* also play a vital role in activating and transferring key proteins in the sub-systems. In our study, expression of *ATG12* and *ATG7* showed up-regulation while *ATG10* expression was down regulated in cancer tissues. Similarly, *Cao et al. (2016)* analyzed 352 tissue microarrays containing cancer and paired adjacent normal tissues and found that *ATG7*, *ATG12* were highly expressed in the GC tissues, and *ATG10* was weakly expressed in GC. Probably because autophagy plays a specific function as a cancer suppressor or tumor promoter, mainly depends on the environment, and its activity will change with the development

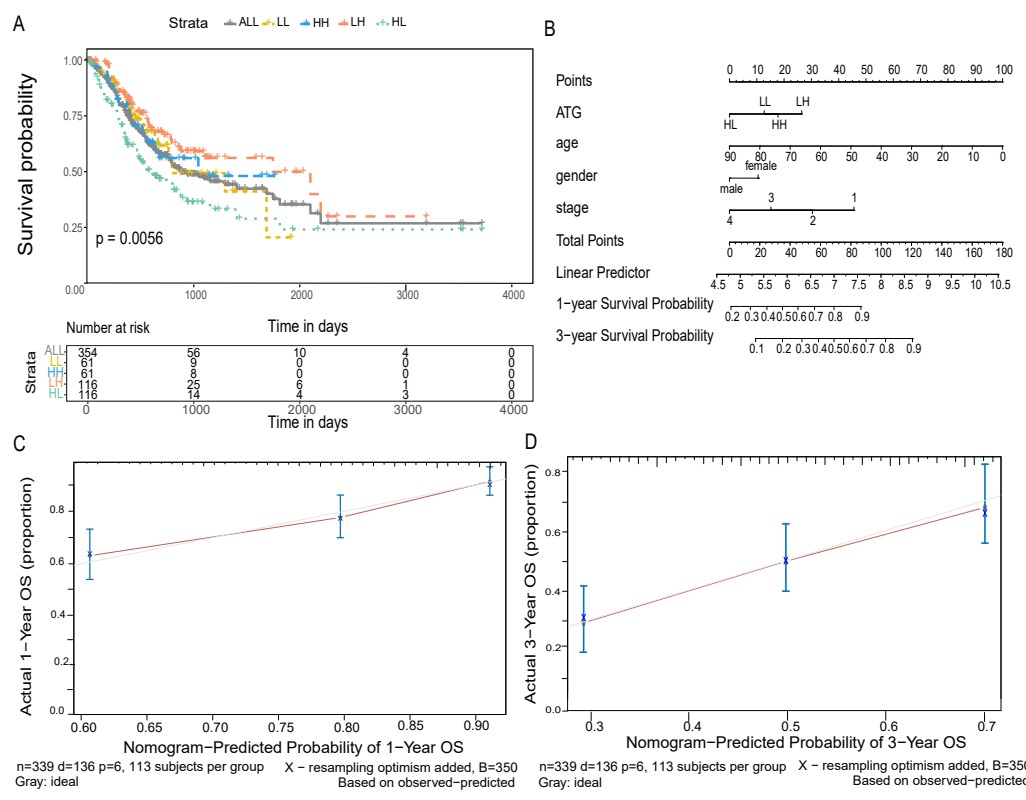

**Figure 6  Joint predictive the patients prognosis using ATG4D and MAP1LC3C.** (A) The Kaplan–Meier plot for the four groups. (B) The nomogram for indicating the one- and three-year overall survival in patients with gastric cancer. (C) The calibration curve of nomogram for predicting overall survival at one year. (D) The calibration curve of nomogram for predicting overall survival at three year.

of the tumor (*Amirfallah et al., 2019*). As for *ATG* genes of other functional units of ubiquitin-like conjugating system, some studies (*Lebovitz et al., 2015*; *Su et al., 2019*) found that *GABARAPL1* transcripts were less abundant in breast, prostate, liver and non-small cell lung cancers than matched normal controls, indicating that *GABARAPL1* may be a tumor suppressor. While *ATG16L2* transcripts increased in kidney cancer. As a high risk gene, its high expression is associated with poor prognosis (*Wan et al., 2019*). The high expression of Ambra1 is beneficial to cell survival (*Sun et al., 2018*). Falasca compared 26 prostate adenocarcinoma and 12 normal specimens by immunohistochemistry and observed that *AMBRA1* was highly expressed in prostate cancer (*Falasca et al., 2015*). The expression trend of those genes was consistent with our results in GC. The above results indicate that the ubiquitin-like conjugated system plays a key role in the occurrence of GC, and its mechanism deserves further study.

It has been reported that autophagy was associated with the invasion, migration as well as implantation metastasis of cancer. Therefore, we further analyzed the relationship between *ATG* genes and GC TNM staging, and verified the differential genes using GSE62254 to improve the accuracy. After verification, *ATG9A*, *ATG2A* and *ATG4D* were found to be associated with T stage. Among these genes, *ATG9A* was previously reported to be

associated with T stage in non-metastatic renal cell carcinoma (*Tang et al., 2013*). *ATG4D* affects the biological behavior of prostate cancer by regulating the activity of androgen receptor (*Hu et al., 2020*). Besides, all of these significant differences were observed at early T stage, which suggested that autophagy may play its role mainly at the early stage of GC progression. By analyzing the expression of *ATG* genes both in TCGA and GSE62254, the results showed that *VMP1* and *ATG4A* were over-expressed in patients with lymph node metastasis. Similarly, Yang et al. found that the expression of *ATG4A* was associated with lymph node metastasis in 110 GC patients (*Yang et al., 2016*). *VMP1* was reported to promote Kras G12D-mediated pancreatic cancer initiation and facilitate lymph node metastasis (*Loncle et al., 2016*). In addition, *ATG4D* and *ATG14* were observed to be associated with overall TNM stage according to our analysis. *ATG14* was up-regulated while *ATG4D* was down-regulated in GC of stage III-IV, which suggested that the two genes may have the opposite effect in GC progression. It has been reported that the low expression of *ATG4D* was associated with Colorectal Cancer of III stage (*Gil et al., 2018*). Interestingly, significant relation was observed between *ATG4* isoforms and all the three clinicopathologic variables, that *ATG4D* was associated with TNM and T stage, and *ATG4A* showed difference in N stage. As *ATG4* activity is essential and highly specific to autophagy, it may be a prospective autophagy-specific target for GC therapy.

Previous investigations have also indicated the predictive role of autophagy pathway genes in prognosis of various types of cancers. Here we analyzed all the *ATG* genes using TCGA dataset to assess their prognostic values in GC. *ATG4D* and *MAP1LC3C* were confirmed to be statistically significant in multivariate survival analysis. The expression of *ATG4D* and *MAP1LC3C* is low in colorectal cancer, and *ATG4D* is related to the poor prognosis of pancreatic cancer (*Hu et al., 2020*). The high expression of *ATG4D* and the low expression of *MAP1LC3C* may indicate the poor survival of gastric patients. Furthermore, we developed a nomogram according to the joint expression of *ATG4D* and *MAP1LC3C* along with other clinicopathological parameters. The group of HL showed poor survival while the group of LH indicated favorable prognosis. In the internal validation set, the calibration plot showed that the predicted 3-year and 5-year overall survival were in correspondence with the actual survival estimated by the Kaplan–Meier method. *MAP1LC3C* is a member of the LC3 family of proteins and a key structural component of the autophagosome that undergoes processing by members of the *ATG4* family (*Costa et al., 2016*). These two functionally related genes together may have synergistic effect in GC prognosis. For the first time, our study formulated an *ATG*-based nomogram that could predict outcome of GC with a better accuracy.

On the basis of the above results, we found that *ATG4* and *ATG8*, members of *ATG8/LC3* system, were associated with both the occurrence and prognosis of GC in our study. *ATG4* was up-regulated in cancer and was associated with poor GC survival. The over-expression of *ATG8* was observed in normal tissues and involved with favorable prognosis of GC. *ATG8/LC3* is essential for autophagosome biogenesis and it also functions as an adaptor protein for selective autophagy (*Lee & Lee, 2016*). At the same time, it is also widely used as a marker of autophagic vacuoles (*Mareninova et al., 2020*).Dysregulation of *ATG8/LC3* proteins may contribute to pathogenic effects during progression of autophagy-associated

human diseases. Our results indicated that the *ATG8/LC3* system may play an important role in development and clinical outcome of GC. Elucidation of alterations in specific *ATG8/LC3* forms in GC could provide insights for the discovery, diagnosis, or targeting of this high-mortality disease.

In conclusion, our study provided a comprehensive illustration of *ATG* genes expression characteristics in GC. Abnormal expressions of *ATG* genes were observed to be significantly involved in the whole process of GC occurrence, progression and prognosis. Specially, the *ULK* system, such as *ATG4* family and *ATG8/LC3*, may serve as valuable biomarkers to indicate gastric carcinogenesis and prognosis. Considering the underlying important roles of *ATG* genes in gastric carcinogenesis and progression, future molecular experiments concerning the functions and mechanisms of *ATG* genes may generate promising significance in GC development and treatment.

## CONCLUSIONS

Our study provided a comprehensive illustration of *ATG* genes expression characteristics in GC. Abnormal expressions of the ubiquitin-like conjugated system in *ATG* genes plays a key role in the occurrence of GC. *ATG8/LC3* sub-system may play an important role in development and clinical outcome of GC. In the future, it is necessary to further elucidate the alterations of specific *ATG8/LC3* forms in order to provide insights for the discovery, diagnosis, or targeting for GC.

### Funding
This work was supported by the National Key R&D Program, Grand 2018YFC1311600 and the Liaoning Provence Key R&D Program (Grant 2020JH2/10300063). The funders had no role in study design, data collection and analysis, decision to publish, or preparation of the manuscript.

### Grant Disclosures
The following grant information was disclosed by the authors:
National Key R&D Program: 2018YFC1311600.
Liaoning Provence Key R&D Program: 2020JH2/10300063.

### Competing Interests
The authors declare there are no competing interests.

### Author Contributions
- Mengya Wang conceived and designed the experiments, performed the experiments, analyzed the data, prepared figures and/or tables, and approved the final draft.
- Jingjing Jing performed the experiments, prepared figures and/or tables, and approved the final draft.
- Hao Li and Jingwei Liu analyzed the data, prepared figures and/or tables, and approved the final draft.

- Yuan Yuan and Liping Sun conceived and designed the experiments, authored or reviewed drafts of the paper, and approved the final draft.

## Data Availability

Data is available at Oncomine: https://www.oncomine.org/resource/login.html.

Data is also available at TCGA: TCGA-STAD and at NCBI GEO: GSE62254.

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
