# Peer review of "The expression characteristics and prognostic roles of autophagy-related genes in gastric cancer"

_PeerJ, doi:10.7717/peerj.10814_

## Round 0.1 · original submission · Major Revisions

Please address critiques of reviewers and revise manuscript accordingly.

Reviewer 1 ·

Basic reporting

no comment

Experimental design

no comment

Validity of the findings

no comment

Additional comments

This is an interesting manuscript however, it must be improved.


ATG is not an abbreviation for "autophagy-related genes" hence using ATGs abbreviation is incorrect. Please correct it according to: Look people, “Atg” is an abbreviation for “autophagy-related.” That’s it. Daniel J. Klionsky. Autophagy. 2012 Sep 1; 8(9): 1281–1282. doi: 10.4161/auto.21812 PMCID: PMC3442874.
Moreover, all gene symbols should be italicized.
Please correct ATG8/LC3 as well.
In the introduction section lack of information about gastric cancer (GC). Please include epidemiology, risk factors, and classification of GC into the manuscript.
Line 96 „different histological types” what types were taken into consideration?
Line 160 HR please explain the abbreviation
Line 196 is: "Adjacent" please correct to "adjacent"
Line 205 is: "Some" please correct to "some"
Line 212 is: "AMBAR1" please correct to "AMBRA1"
Line 234 with III stage but what type of cancer?
Line 237 Atg4b – what do you mean? A mouse system?

Reviewer 2 ·

Basic reporting

Minor comments:
Articles must revise with grammatical and typical errors eg, line 171(-year).
Improvement in figure qualities will strength the article.

Experimental design

Title: The expression characteristics and prognostic roles of autophagy-related genes in gastric cancer

Authors characterized the expression of different genes which responsible for autophagy mechanism in gastric cancer. The study design and usage of bioinformatics tools is appropriate for this study.

Validity of the findings

The study reports the upregulation of ATG8/LC3 has direct effect on gastric cancer. Overall the study design, material and method selection is supporting the concept of study.

Reviewer 3 ·

Basic reporting

Clear outline of the paper with a sound description of aims and objectives. Citations are relevant and recent.

Experimental design

Authors have employed a reasonable experimental design. The subject selection us clear. Here are my comments on the methods section:
1) Page 90: Please cite a relevant reference for Pathvisio. Also, please mention any further details like version, URL, settings used etc.
2) Authors have cited reference for Oncomine and TCGA dataset. Please provide any relevant details like version, URL etc.
3) Lines 102-103: Authors write – “The patients’ TNM stage, T, N and M were considered as the clinical parameters”. Please explain the criteria for deciding these clinical parameters and/or cite a relevant reference.

Validity of the findings

This is an interesting and informative study; and the findings would contribute to the progression of knowledge in the field. The manuscript would gain much if authors provide any biochemical validation of the results pertaining to the differentially expressed genes in GC vs normal cells by techniques like RT-qPCR, immunoblot etc.

Additional comments

The manuscript submitted by Wang. Jing et al., presents a study on identification of the role of autophagy related genes (ATGs) in the occurrence and development of gastric cancer (GC). Using previously published datasets, authors studied the expression of ATGs in cancer and normal tissues. Authors identified several ATGs associated with either TNM stage, T stage or lymph node metastasis stage. Authors conclude that Alternations in the expression of the ubiquitin-like conjugated system in ATGs might play a key role in GC occurrence. Overall, the manuscript is well written and informative. Here are a few remarks that need to addressed.
1) Authors need to include more introductory literature on the already published literature on ATGs’ association with cancer, especially the GC cases. I would also recommend if authors include molecular details behind the association.
2) It would be helpful if authors summarize their findings in the form of a schematic model.
3) The discussion section is long and hence can be shortened.

---

## Round 0.2 · accepted · Accept

All critiques were addressed and the manuscript was amended accordingly. Therefore, I am pleased to accept your manuscript for publication in PeerJ.

Reviewer 3 ·

Basic reporting

The manuscript meets basic criteria of reporting and publication.

Experimental design

Experimental design employed here is scientifically sound and the revised version provides all the relevant information on it.

Validity of the findings

The findings are novel and provide insights into the biology of autophagy-related genes in gastric cancer.

Additional comments

In the revised version, authors have addressed my concerns. I recommend this manuscript for publication in PeerJ.